# Gastrointestinal and Hepatological Manifestations in Severe Acute Respiratory Syndrome Coronavirus 2 Infection: Results from the Major COVID Hospital in Serbia

**DOI:** 10.3390/microorganisms12010027

**Published:** 2023-12-22

**Authors:** Dragana Mijac, Samir Vucelj, Kristina Todorovic, Marko Vojnovic, Biljana Milicic, Snezana Lukic, Branka Filipovic, Marija Marjanovic Haljilji, Dusan Popovic, Tatjana Adzic Vukicevic

**Affiliations:** 1Faculty of Medicine, University of Belgrade, Dr Subotica Starijeg 8, 11000 Belgrade, Serbia; lukic.snezana@gmail.com (S.L.); branka.filipovic3@gmail.com (B.F.); dr.dusan.popovic@gmail.com (D.P.); adzic_tatjana@yahoo.com (T.A.V.); 2Clinic for Gastroenterology and Hepatology, University Clinical Center of Serbia, Koste Todorovica No 2, 11000 Belgrade, Serbia; marko.vojna@gmail.com; 3Department of Internal Medicine, General Hospital Novi Pazar, Generala Zivkovica 1, 36300 Novi Pazar, Serbia; vucelj.samir@hotmail.com; 4Department of Internal Medicine, General Hospital Jagodina, Karadjordjeva 4, 35000 Jagodina, Serbia; dr.kristinatodorovic@gmail.com; 5Faculty of Dental Medicine, University of Belgrade, Rankeova 4, 11000 Belgrade, Serbia; milicic.biljana85@gmail.com; 6Department of Gastroenterology, Clinical and Hospital Center “Dr Dragisa Misovic–Dedinje”, Heroja Milana Tepica 1, 11020 Belgrade, Serbia; maja-s-92@hotmail.com; 7Clinic for Pulmology, University Clinical Center of Serbia, Dr Koste Todorovica No 6, 11000 Belgrade, Serbia

**Keywords:** COVID-19, SARS-CoV-2, gastrointestinal symptoms, liver damage

## Abstract

The coronavirus disease of 2019 (COVID-19), caused by severe acute respiratory syndrome coronavirus type 2 (SARS-CoV-2), includes a clinical spectrum of diseases from mild to severe progressive pneumonia, which has affected and still affects the human population worldwide. Most commonly, it is presented by respiratory symptoms, but studies have shown that about 50% of patients with SARS-CoV-2 infection have at least one gastrointestinal symptom (GI), predominantly nausea, diarrhea, vomiting, or loss of appetite. In addition, abnormal liver functional tests are commonly present in the SARS-CoV-2 virus. The aim of our study was to examine the GI and hepatic manifestations of COVID-19 in patients hospitalized due to COVID-19 pneumonia in “COVID hospital Batajnica”, University Clinical Center of Serbia in Belgrade. The study included 498 consecutive patients, and the data was obtained from the patient’s electronic medical history. GI symptoms included nausea, vomiting, diarrhea, and anorexia. Collected laboratory values included baseline and peak values of blood count, inflammatory parameters, liver function tests, renal function tests, and cardiac enzyme tests. The results have shown that GI symptoms occurred in 26% of cases at diagnosis, which indicates the great susceptibility of the GI system to SARS-CoV-2. There was a high risk of liver injury in patients with COVID-19 pneumonia (>60%). The level of AST is more often increased compared to ALT, which is different from other virus-induced liver lesions and may be a useful indicator of SARS-CoV-2 infection. Further research should focus on the causes of liver damage in SARS-CoV-2 virus and the impact on treatment and outcome of COVID-19 disease.

## 1. Introduction

The severe acute respiratory syndrome coronavirus-2 (SARS-CoV-2) started in 2019 in Wuhan, China, and was soon identified for the first time on 7 January 2020, rapidly becoming a global pandemic. SARS-CoV-2 is an RNA virus responsible for Coronavirus-19 disease (COVID-19). COVID-19 may have a spectrum of clinical manifestations ranging from mild symptoms or asymptomatic disease to rapidly progressive acute respiratory distress syndrome (ARDS) and life-threatening conditions. According to the World Health Organization (WHO) COVID-19 situation report, 539,893,858 cases of COVID-19 had been confirmed by June 2022, and 6,324,112 people had died, which indicates that the mortality from the disease was 1.17% [1].

According to published data, COVID-19 is manifested predominantly by respiratory symptoms, with fever and cough being the most common symptoms. However, the incidence of other clinical manifestations differs in published reports, showing that gastrointestinal (GI) symptoms and abnormal liver biochemistry are reported commonly [2,3]. According to recent publications, GI manifestations include diarrhea, nausea, vomiting, acid regurgitation, loss of appetite, and abdominal pain, while the prevalence of GI symptoms varies widely from 2% to 57% in different studies [4,5]. It is shown that about 50% of patients with SARS-CoV-2 infection have at least one GI symptom. About 2–10% of patients with COVID-19 had reported diarrhea [6].

The SARS-CoV-2 virus enters cells through protein receptors, specifically angiotensin-converting enzyme 2 (ACE 2) receptors. These receptors are highly expressed in respiratory cells but are also expressed in the enterocytes, mostly of the ileum and colon. Thus, GI manifestations in a considerable number of affected individuals may be due to the SARS-CoV-2 tropism for this peptidase angiotensin receptor. Moreover, intestinal biopsy specimens from patients with COVID-19 detected SARS-CoV-2 nucleic acid, confirming active viral replication in the GI tract [7]. Thus, SARS-CoV-2 RNA can be detected in the stool of patients with GI manifestations of COVID-19, indicating the possibility that the fecal-oral route of transmission is also possible [8,9]. Also, many studies confirmed that the liver can be affected by SARS-CoV-2 virus infection, and some authors showed that abnormal liver biochemistry correlates with the severity of the disease. 

As the biggest COVID hospital in Serbia, we seek to contribute to worldwide data that outlines the GI and hepatic manifestations of COVID-19 in Serbia.

The aim of our study was to examine the GI and hepatic manifestations of COVID-19 in patients hospitalized due to COVID-19 pneumonia in the major COVID Hospital in Serbia “COVID Hospital Batajnica”, University Clinical Center of Serbia, Belgrade. 

## 2. Material and Methods

### 2.1. Study Design and Data Source

This descriptive, cross-sectional study was conducted on a cohort of COVID-19 patients who were admitted to the “COVID Hospital Batajnica” University Clinical Center of Serbia, in Belgrade. 

The study was approved by the Institutional Board of “COVID Hospital Batajnica” and the Ethics Comity, University Clinical Center of Serbia (No 808/7). We included consecutive symptomatic patients requiring hospital admission with a confirmed diagnosis of COVID-19 infection on nasopharyngeal PCR testing for SARS-CoV-2 infection from 1 November to 15 December 2021. Patients who did not require hospitalization were managed on an ambulatory basis, and these data were unavailable and thus not considered for the purposes of this study. In order to reduce bias regarding GI manifestations of COVID-19 and abnormal liver biochemistry of other etiology, we applied the following exclusion criteria for entry into the study: patients with confirmed Clostridium difficile infection or other distinctive colitis (i.e., inflammatory bowel disease), patients with acute gastrointestinal bleeding and patients with previously known chronic liver disease of any etiology. Also, patients with severe COVID-19 disease (critically ill patients requiring invasive mechanical ventilation and/or patients with multiorgan failure) were not included in the study due to potential bias. We also excluded patients who were clinically suspect, but PCR SASR-CoV-2 nasopharyngeal swab results were negative. 

The study included 498 consecutive patients with confirmed SARS-CoV-2 infection who were hospitalized in “COVID Hospital Batajnica” due to COVID-19 pneumonia according to the WHO guidelines.

Data were collected as comprehensively as possible through a combination of electronic medical systems and through communication with attending doctors to fill in the missing data. All data was separately collected by two authors (S.V. and K.T.).

We extracted demographic data, clinical characteristics including respiratory symptoms, hemodynamic parameters, and digestive symptoms on admission, comorbidities, a complete panel of routine laboratory tests, and chest X-ray or computerized tomography (CT). 

For all included patients the following data was collected: sex, age, comorbidities, pulmonary X-ray and/or chest computed tomography (CT) scan, hemodynamic parameters including oxygen saturation rate and oxygen therapy if needed, laboratory analyses (C reactive protein (CRP), erythrocyte sedimentation rate, procalcitonin (PCT) levels, ferritin blood test, D-dimer levels, liver functional tests, renal function tests, and cardiac enzymes tests). Collected laboratory values regarding liver biochemistry included baseline and peak values of bilirubin, alanine aminotransferase (ALT), aspartate aminotransferase (AST), alkaline phosphatase (AP), and gamma-glutamyl transpeptidase (GGT). Laboratory values of liver enzymes were marked as normal, marginally elevated (<2 × above the upper normal value), slightly elevated (2–5 × above the upper normal value), moderately elevated (5–15 × above the upper normal value) and severely elevated (>15 × above the upper normal values). The categories of laboratory values were based on the guides of the European Association for the Study of the Liver (EASL) and the American Association for the Study of Liver Diseases (AASLD). Gastrointestinal symptoms included nausea, vomiting, diarrhea, and anorexia. Comorbidities included chronic lung disease, cardiovascular diseases (CVD), diabetes mellitus (DM), chronic kidney disease, cerebrovascular disease, malignancy, and neurological or psychiatric disorders. Vaccinated status was obtained for each patient.

### 2.2. Statistical Analysis

Descriptive statistics were calculated for demographic characteristics and other parameters and were presented as frequencies and proportions for discrete measures. All our data were categorical and analyzed using the Pearson chi-square test. For statistical analysis differences between groups with and without gastrointestinal symptoms (dependent variable) or with and without elevated liver chemistry (dependent variable), univariable and multivariable logistic regression methods were used analysis to calculate odds ratios (OR) and their 95% confidence intervals (CI). The association of gastrointestinal symptoms and abnormal liver chemistry with observed risk factors (independent variables) was examined by use of an univariable model of logistic regression. All test variables with a statistical significance of *p* < 0.05 in the univariable model were included in the multivariable model. Significant parameters in the multivariate model were considered independent predictors of difference in primary outcome variables: gastrointestinal symptoms and elevated liver chemistry. The established level of significance was 0.05. Statistical analysis was performed using the IBP SPSS Statistics v28 (Statistical Package for Social Sciences, SPSS Inc., Chicago, IL, USA).

## 3. Results

The study included 498 patients, 268 male and 230 female. The average age of the patients was 70.6 ± 13.38 (71; 23–99). Most of the patients were elderly population (>65 years of age) with one or more comorbidities (73.4%). The total percentage of vaccinated patients against the SARS-CoV-2 virus was 37.8%. 

Demographic characteristics of patients with COVID-19 are presented in Table 1.

Gastrointestinal symptoms (at least one) were experienced by 128 patients (25.7%). Most of the patients had poor appetite (65 of them; 13.1%), followed by patients with diarrhea (47; 9.4%). Nausea was experienced by 19 patients (3.8%), and vomiting was present in 17 patients (3.4%). The values of routine laboratory tests, including blood counts, parameters of inflammation, and abnormal liver biochemistry, are shown in Table 1.

Regarding liver biochemistry, AST was more often increased compared to ALT (AST 67%, ALT 56.5%). Mild elevation of AST was present in 90% of patients, while mild elevation of ALT was observed in over 95% of patients. AST median was 91 (14–501), ALT median 98.6 (6–425); elevated GGT was observed in 80% of patients with a median of 288 (12–1388), and elevated AP was in 18.3% of patients.

Gastrointestinal manifestations and abnormal liver chemistry regarding vaccination against the SARS-CoV-2 virus are shown in Table 2.

Previous vaccination against the SARS-CoV-2 virus did not affect the presence of GI manifestations and/or abnormal liver chemistry in patients with COVID-19.

The prevalence of various GI symptoms and elevated liver chemistry according to gender and vaccination against the SARS-CoV-2 virus is shown in Table 2.

Considering gender, GI symptoms were more common in women than in men (31.3% vs. 29%), while abnormal liver function tests were more common in men than in women, with a statistically significant difference (90.2% vs. 81%, *p* < 0.01). 

Gastrointestinal manifestations are more common in female patients than in males (31.3% vs. 20.9%, *p* < 0.01).

The prevalence of various GI symptoms and elevated liver chemistry according to age and vaccination against the SARS-CoV-2 virus is shown in Table 3.

Men older than 45 years had higher liver enzymes more often. Independent predictors for elevated liver enzymes were male gender, while GI symptoms were a negative predictor for abnormal liver function tests.

The prevalence of GI manifestations and abnormal liver biochemistry regarding the presence of any comorbidities are shown in Table 4.

Gastrointestinal manifestations were more frequent in patients with one or more comorbidities than in patients who did not have any comorbidities (56.1% vs. 40.3%, *p* < 0.05), while the presence of comorbidities did not affect the liver biochemistry profile (NS).

Gastrointestinal manifestations were more frequent in patients with CVD than in patients who did not have CVD (58.7% vs. 42.9%, *p* < 0.05), while the presence of CVD did not affect the liver biochemistry tests (NS). The prevalence of GI manifestations and abnormal liver biochemistry regarding the presence of CVD are shown in Appendix A.

Gastrointestinal manifestations were more frequent in patients with DM than in patients who did not have DM (64.9% vs. 48.3%, *p* < 0.05), while the presence of DM did not affect the liver biochemistry tests (NS). The prevalence of GI manifestations and abnormal liver biochemistry regarding the presence of DM are shown in Appendix A.

The presence of chronic renal disease did not affect the liver biochemistry tests (NS) nor the presence of GI symptoms (NS). The prevalence of GI manifestations and abnormal liver biochemistry regarding the presence of chronic kidney disease are shown in Appendix A.

Predictors of GI manifestation in patients with COVID-19 are shown in Table 5.

Independent predictors for GI symptoms were female gender and leucopenia, while lower ferritin level, DM, and CVD did not show a statistically significant influence on the presence of GI symptoms in the multivariate model.

Predictors of elevated liver biochemistry in patients with COVID-19 are shown in Table 6.

Independent predictors for elevated liver enzymes were diarrhea and loss of appetite as negative predictors for abnormal liver biochemistry (patients with diarrhea and loss of appetite had more frequently normal liver enzymes) and previous chronic respiratory disease. Interestingly, patients with previous chronic respiratory disease had more often normal liver enzyme tests than patients without previous history of pulmonary diseases. This result has been considered as a coincidence. Vaccination did not influence laboratory parameters of liver enzyme tests.

## 4. Discussion

In our study, 73.4% of patients had one or more comorbidities, which can be explained by the fact that most of the patients were elderly population. Of all comorbidities, CVD is the most prevalent in the percentage of 27.9%, followed by DM with a participation of 11.4%, and pulmonary diseases were in the third place with a frequency of 5%, as shown by previous studies [10]. Only 37.8% of our hospitalized patients were vaccinated, and that was statistically significantly higher in patients over 65 years of age. The reason for this is that during the pandemic, all people over the age of 65 were the first to be vaccinated. Men were vaccinated statistically significantly more often than women. Previous vaccination against the SARS-CoV-2 virus did not affect the presence of GI manifestations and/or abnormal liver chemistry in patients with COVID-19.

Regarding gender, GI symptoms occurred significantly more often in the female than in the male population in our study (31.3% vs. 29%), while in the meta-analysis conducted by Zaman et al. GI symptoms were significantly more often reported in men than women [11].

At least one GI symptom occurred in 25.7% of patients, with loss of appetite being the most common manifestation (13.1%), followed by diarrhea, nausea, and vomiting, respectively. These results are in accordance with those reported throughout the literature, indicating high susceptibility of the GI tract to SARS-CoV-2. In the study conducted by H. Zhang et al., similar results were demonstrated, with the same predominance of the most common symptoms: loss of appetite (56.7%), diarrhea (37.8%), and nausea (16.5%). Fang et al. showed a somewhat higher incidence of GI symptoms—with 79.1% of all patients experiencing GI symptoms [12,13]. By reviewing the literature, it could be found that diarrhea has participated in more severe forms of the disease [14]. Han et al. reported that 20% of their patients had diarrhea as an initial symptom and that, in other cases, it occurred up to 10 days after the onset of respiratory symptoms. Most reported patients had a mild form of diarrhea [15]. Luo et al. showed a high incidence of nausea and vomiting, whereas the incidence of nausea in most other studies was lower. A higher incidence of nausea was associated with more severe forms of the disease [16]. In the study by Wang et al., abdominal pain occurred less frequently than other GI complaints but was more common in those patients who were hospitalized in intensive care units [17]. Seelinger et al. described seven cases in which patients underwent emergency surgery for acute abdomen [18]. In a meta-analysis of 60 studies, 26.8% of patients with COVID-19 had anorexia as the most common symptom, making enteral feeding difficult for these patients, as demonstrated in our results. In the previously mentioned analysis, some patients who later developed refractory pneumonia had pronounced anorexia on admission [19]. In our study, independent predictors for the occurrence of GI symptoms were female gender, comorbidities (DM), leukopenia, and lower values of ferritin. A statistically significant difference was observed in average ferritin values between subjects with and without the onset of GI symptoms during SARS-CoV-2 infection. Subjects without GI symptoms had statistically significantly lower ferritin values than subjects with GI symptoms. So far, it has been shown that higher levels of ferritin are predictors of severe forms of COVID-19 and worse clinical outcomes [20,21]. Although this could lead us to the possibility of thinking that patients with GI symptoms would have worse clinical outcomes, so far, studies have given controversial results. Large meta-analyses have concluded that, despite being very common, the presence of GI symptoms does not overall affect COVID-19 severity and mortality rates [11,22]. A study conducted by Milano et al. has shown that, despite the presence of liver damage, GI involvement did not affect the COVID-19 overall outcome, duration of hospitalization, need for non-invasive ventilation (NIV), or mortality [23]. On the other hand, a meta-analysis conducted by Mao et al. demonstrated that patients with GI presentation of COVID-19 tended to have poorer clinical outcomes [4]. 

Regarding liver biochemistry, mild elevation of AST was present in 90% of patients, while mild elevation of ALT was observed in over 95% of patients. Men over 45 years of age had elevated liver enzymes more often, as did the population over 65 years of age, which can be explained by the fact that metabolic liver diseases are more common in men, as well as polypharmacy in patients over 65 years old [24]. Liver enzymes were frequently elevated in patients with COVID-19 pneumonia in over 60% of cases. The level of AST was more often increased compared to ALT (AST 67%, ALT 57%), which is distinct from other viral-induced liver injuries and may be a useful indicator of SARS-CoV-2 infection. This may be owing to the wider parenchymal distribution of AST (skeletal muscle, cardiac, kidney, and lung tissue), which supports multiorgan injury seen in COVID-19, which was confirmed in a large meta-analysis [25]. Elevated AST values indicated a worse prognostic indicator [26]. Also, elevated values of AST predominantly in men can be considered in the context of previously unrecognized liver damage of ethylic etiology, which is more common in the general population of men. A total of 80% of patients with abnormal liver function tests had elevated GGT values. It is assumed that one of the main reasons for this is the increased intake of various medications that were used during the patient’s treatment. In a multicenter retrospective study by Kasapoglu et al., it was shown that serum GGT, creatinine, and d-dimer level have an important predictive role and indicate the need for hospitalization of patients in the intensive care unit as well as higher mortality [27]. In this study, independent predictors for elevated liver enzymes were male sex, while GI symptoms were a negative predictor for abnormal liver function tests. 

Considering comorbidities, our research found that GI manifestations were more frequent in patients with CVD than in patients who did not have CVD (58.7% vs. 42.9%, *p* < 0.05). Also, a statistically significant occurrence of GI symptoms was found in the group of patients with DM when compared to patients without DM (64.9% vs. 48.3%, *p* < 0.05). The presence of CVD and/or DM did not affect the liver biochemistry tests (NS). The presence of chronic kidney disease did not affect the liver biochemistry tests (NS) nor the presence of GI symptoms (NS). 

It is well documented that diabetes mellitus, CVD (hypertension), chronic kidney diseases, and malignancies were independently associated with worse outcomes and mortality in COVID-19 patients, but we have not found data regarding the association between GI manifestations and comorbidities in patients with COVID-19 disease, so far [28]. 

In our study, through logistic regression, we represent adjusted OR, the logarithmic form of regression coefficients (β) (using the natural base represented by “e”). Some input variables in our model are highly correlated with one another (known as multicollinearity), so the effect of each on the regression model becomes less precise. We considered a model where comorbidity, CDV, and DM were used as input variables to predict the risk of developing GI manifestations in patients with COVID-19. Because comorbidity includes both variables, CVD and DM. In our regression model, we include CVD and DM. The same situation occurred in the regression model where the dependent variable was elevated liver biochemistry in patients with COVID-19. Predictor “GI symptoms” structurally collinear with both “loss of appetite” and “diarrhea”. In multivariate models, we include only loss of appetite and diarrhea. Finally, we used adjusted logistic regression in both situations. We include covariates (confounders): age, sex, and vaccination as input variables. The aim of this study was to examine the GI and hepatic manifestations of COVID-19 in patients hospitalized due to COVID-19 pneumonia. We resolved the imbalance in observed risk factors between groups with and without GI manifestation in patients with COVID-19 or between groups with and without elevated liver biochemistry in patients with COVID-19 using adjustments.

It is shown that patients with COVID-19 presenting with GI symptoms had reduced mortality. Furthermore, COVID-19 severity was significantly reduced in patients with GI manifestations when compared to those without GI manifestations [29]. This can be explained by reduced levels of circulating cytokines associated with systemic inflammation and tissue damage in patients with GI symptoms. It has been shown that serum levels of IL-6 and IL-8, which are known to be directly associated with poor survival, are found to be significantly reduced in the circulation of patients with predominant GI symptoms [29]. On the other hand, some authors reported that patients with GI symptoms had a more severe course of COVID-19 with a higher incidence of progression to severe disease and a longer hospital stay. These authors also showed that those patients with predominant GI symptoms had significantly later hospital admission than those with primary respiratory symptoms, and that can be one of the explanations for further delay of adequate therapy and worse prognosis [15,30]. 

However, the potential role of the GI tract in the pathogenesis of SARS-CoV-2 infection needs to be further examined. 

Our study has some limitations regarding the study’s design. This is a cross-sectional study, and there was a lot of uncompleted data due to limited information and bias of both patients and medical professionals involved at the time of intake. In the present study, we found gastrointestinal symptoms (at least one) at a rate of 25.7%. Given a binary outcome occurring in 25.7% of patients, at least 380 participants are necessary to achieve a confidence interval of 0.2 to 0.5 for the overall outcome proportion at a two-tailed significance level of 0.05 and a power of 0.8. Additionally, we enrolled 15% more patients to mitigate potential reductions in sample size or protocol deviations. This precaution ensured that our study remained robust and the findings were credible, but unfortunately, only for overall analysis.

Moreover, we excluded patients with severe COVID-19 infection treated in ICU, who required mechanical ventilation, or developed ARDS, MOF, or acute kidney injury and needed renal replacement therapy due to many different factors that can influence liver injury and different GI symptoms in these most severe groups of patients. 

Regarding liver injury, although there was no obvious bias in the definition of liver injury, we could not exclude the potential effect of medications, sepsis, or other etiology factors on liver biochemistry. Moreover, we could not exclude the effects of other confounding factors, such as age, gender, comorbidities, and complications, on the results of our study. A more detailed profile of specific aspects from published data of different countries should emerge in future studies in order to get more comprehensive answers on this topic.

## 5. Conclusions

In summary, we presented our data from a large single-center analysis of GI and hepatic manifestations of COVID-19 in Serbian patients. We found that COVID-19 is associated with GI symptoms such as diarrhea, nausea, and vomiting in a high proportion of infected patients (26%) at diagnosis, which confirms awareness of the role of GI involvement in SARS-CoV-2 infection. 

There is a high risk of liver injury in Serbian patients with COVID-19 infection with pneumonia, and mildly elevated transaminase levels are very common (>60%). The level of AST is more often increased compared to ALT, which is different from other virus-induced liver injuries and may be a useful indicator of SARS-CoV-2 infection. This may be due to a wider parenchymal distribution of AST (skeletal muscle, heart, kidney, and lung tissue), which supports the thesis of multiorgan pathogenesis in COVID-19. Thus, liver enzyme abnormality is very likely to be multifactorial. Further research should focus on the causes of liver damage in COVID-19 and the impact on treatment and outcome of COVID-19.

We report a large, single-center analysis of the GI and hepatic manifestations of COVID-19 from the major COVID-19 Hospital in Serbia. GI symptoms and abnormal liver biochemistry were common in our cohort and may be clinically useful in stratifying the risk of disease severity and prognosis.

## Figures and Tables

**Table 1 microorganisms-12-00027-t001:** Demographics and anamnestic characteristics of patients included in the study, as well as laboratory analyses and abnormal liver chemistry in patients with COVID-19.

Characteristics	n (%)/X ± SD (Med; Min–Max)
Patients	498
Demographics	
Age	70.6 ± 13.38 (71; 23–99)
Age <45/45–65/>65	33 (6.6%)/112 (22.5%)/353 (70.9%)
Sex (Male/Female)	268 (53.8%)/230 (46.2%)
Vaccination	
Vaccination (Yes/No)	188 (37.8%)/310 (62.2%)
Risk factors	
Comorbidities (all)	171 (73.4%)/62 (26.6%)
Cardiovascular disease (CVD)	139 (27.9%)
Pulmonary disease	26 (5.2%)
Malignancies	9 (1.8%)
Diabetes mellitus (DM)	57 (11.4%)
Neurological/psychiatric disorders	25 (5.0%)
Kidney disease	22 (4.4%)
Oxygen saturation	
SaO_2_	94 ± 4.46 (95; 50–99)
O_2_/L	5.24 ± 6.63 (3; 0–40)
O_2_/L > 5 L/min	93 (18.7%)
Gastrointestinal symptoms	
GI symptoms (all)	128 (25.7%)
Loss of appetite	65 (13.1%)
Nausea	19 (3.8%)
Vomiting	17 (3.4%)
Diarrhea	47 (9.4%)
Laboratory analyses	n (%)/X ± SD (Med; min–max)
Blood count	
Leukocytes	7.49 ± 4.14 (7.05; 0.5–27.6)
Lymphocytes	1.01 ± 0.87 (0.865; 0.1–9.63)
Platelets	200.83 ± 107.463 (180; 23–998)
Hemoglobin	130.48 ± 26.93 (130; 41–246)
Inflammatory parameters	
CRP	351.70 ± 934.76 (110.35; 0.6–10,452)
Fibrinogen	5.64 ± 1.75 (5.7; 1.1–10.8)
Ferritin	733.15 ± 994.79 (321.8; 6.6–6486)
Procalcitonin (PCT)	4.87 ± 46.00 (0.1; 0.01–63.2)
D-dimer	2.91 ± 6.02 (1.36; 0.27–51.6)
LDH	603.87 ± 393.59 (557.5; 23–1641)
Liver chemistry	n (%)/X ± SD (Med; min–max)
AST (U/L) on admission	76.36 ± 75.51 (54; 13–460)
Elevated AST: yes/no	155 (67.1%)/76 (32.9%)
AST: mild/moderate elevation	142 (28.5%)/13 (2.6%)
AST maximum value	91.52 ± 86.95 (66; 14–545)
ALT (U/L) on admission	60.32 ± 55.43 (47; 3–501)
Elevated ALT: yes/no	130 (56.5%)/100 (43.5%)
ALT: mild/moderate/severe elevation	126 (25.3%)/3 (0.6%)/1 (0.2%)
ALT maximum value	98.57 ± 288.27 (64; 6–425)
Bilirubin	79.98 ± 90.16 (45.50; 8.8–483)
Elevated bilirubin: yes/no	45 (90.0%)/5 (10.0%)
GGT	288.18 ± 341.47 (96; 12–1388)
Elevated GGT: yes/no	117 (79.1%)/31 (20.9%)
AP	116.37 ± 188.33 (47; 6–842)
Elevated AP: yes/no	11 (18.3%)/49 (81.7%)
Elevated AST and/or ALT: yes/no	185 (85.5%)/45 (15.5%)
Elevated liver chemistry (any): yes/no	201 (86.3%)/32 (13.7%)

**Table 2 microorganisms-12-00027-t002:** Prevalence of various GI symptoms and elevated liver chemistry according to gender and vaccination against the SARS-CoV-2 virus.

**Characteristics**			**Gender**		**Significance**
			**Male**	**Female**	
Demographics					
Age	<45		20 (7.5%)	13 (5.7%)	*p* = 0.560
	45–65		63 (23.5%)	49 (21.3%)	
	>65		185 (69.0%)	168 (73.0%)	
Vaccination					
Vaccination			113 (42.2%)	75 (32.6%)	*p* = 0.028 *
GI symptoms					
GI symptoms (all)			56 (20.9%)	72 (31.3%)	*p* = 0.008 *
Loss of appetite			30 (11.2%)	35 (15.2%)	*p* = 0.184
Nausea			9 (3.4%)	10 (4.3%)	*p* = 0.565
Vomiting			3 (1.1%)	14 (6.1%)	*p* = 0.002 *
Diarrhea			20 (7.5%)	27 (11.7%)	*p* = 0.104
Elevated liver chemistry			120 (90.2%)	81 (81.0%)	*p* = 0.043 *
**Characteristics**		**Vaccination**			**Significance**
		**No**		**Yes**	
GI symptoms					
GI symptoms		83 (26.8%)		45 (23.9%)	*p* = 0.482
Loss of appetite		42 (13.5%)		23 (12.2%)	*p* = 0.673
Nausea		13 (4.2%)		6 (3.2%)	*p* = 0.571
Vomiting		14 (4.5%)		3 (1.6%)	*p* = 0.082
Diarrhea		31 (10.0%)		16 (8.5%)	*p* = 0.582
Liver chemistry					
Elevated AST		101 (70.1%)		54 (62.1%)	*p* = 0.206
Elevated ALT		85 (59.0%)		45 (52.3%)	*p* = 0.321
Elevated bilirubin		30 (93.8%)		15 (83.32%)	*p* = 0.239
Elevated GGT		72 (75.8%)		45 (84.9%)	*p* = 0.191
Elevated AP		8 (20.5%)		3 (14.3%)	*p* = 0.552
Elevated AST and/or ALT		120 (83.3%)		65 (75.6%)	*p* = 0.152
Elevated: any chemistry		128 (88.3%)		73 (83.0%)	*p* = 0.253

* Statistically significant.

**Table 3 microorganisms-12-00027-t003:** Prevalence of various GI symptoms and elevated liver chemistry according to age and vaccination against the SARS-CoV-2 virus.

Characteristics	Age (Years)			Significance
	<45	45–65	>65	
Vaccination				
Vaccination	8 (24.2%)	20 (17.9%)	160 (45.3%)	*p* = 0.001 *
GI symptoms				
GI symptoms (all)	10 (30.3%)	20 (17.9%)	98 (27.8%)	*p* = 0.093
Loss of appetite	3 (9.1%)	11 (9.8%)	51 (14.4%)	*p* = 0.351
Nausea	2 (6.1%)	3 (2.7%)	14 (4.0%)	*p* = 0.647
Vomiting	5 (15.2%)	1 (0.9%)	11 (3.1%)	*p* = 0.001 *
Diarrhea	5 (15.2%)	6 (5.4%)	36 (10.2%)	*p* = 0.159
Liver chemistry				
Elevated AST	15 (68.2%)	36 (75.0%)	104 (64.6%)	*p* = 0.401
Elevated ALT	12 (54.5%)	35 (72.9%)	83 (51.9%)	*p* = 0.035 *
Elevated AST and/or ALT	17 (77.3%)	45 (93.8%)	123 (76.9%)	*p* = 0.033 *

* Statistically significant.

**Table 4 microorganisms-12-00027-t004:** Presence of any comorbidities and prevalence of GI symptoms and abnormal liver biochemistry in patients with COVID-19.

Characteristics	Any Comorbidities		Significance
	No	Yes	
GI symptoms			
GI symptoms	25 (40.3%)	96 (56.1%)	*p* = 0.033 *
Loss of appetite	15 (24.2%)	47 (27.5%)	*p* = 0.615
Nausea	3 (4.8%)	16 (9.4%)	*p* = 0.265
Vomiting	2 (3.2%)	15 (8.8%)	*p* = 0.150
Diarrhea	8 (12.9%)	35 (20.5%)	*p* = 0.188
Liver chemistry			
Elevated AST	41 (69.5%)	113 (66.5%)	*p* = 0.670
Elevated ALT	37(62.7%)	91 (53.8%)	*p* = 0.237
Elevated bilirubin	13 (100.0%)	31 (86.1%)	*p* = 0.156
Elevated GGT	34 (85.0%)	82 (77.4%)	*p* = 0.308
Elevated AP	2 (20.0%)	9 (18.4%)	*p* = 0.904
Elevated AST and/or ALT	48 (81.4%)	135 (79.9%)	*p* = 0.806
Elevated: any chemistry	53 (89.8%)	146 (85.9%)	*p* = 0.439

* Statistically significant.

**Table 5 microorganisms-12-00027-t005:** Predictors of GI manifestation in patients with COVID-19.

Covariate	Univariate Regression Analyses		Multivariate Regression	
	Exp (B) 95%CI	Significance	Exp (B) 95%CI	Significance
Age	1.190 (0.840–1.686)	*p* = 0.328	/	/
Sex	1.725 (1.150–2.588)	*p* = 0.008 *	3.417 (1.868–6.249)	*p* = 0.000 *
Vaccination	0.861 (0.566–1.308)	*p* = 0.483	/	/
SpO_2_	1.020 (0.961–1.082)	*p* = 0.522	/	/
O_2_/L	1.051 (0.621–1.779)	*p* = 0.335	/	/
Comorbidity	1.894 (1.050–3.419)	*p* = 0.034 *	0.825 (0.301–2.260)	*p* = 0.708
CVD	1.895 (1.109–3.238)	*p* = 0.019 *	1.716 (0.907–3.246)	*p* = 0.097
Respiratory disease	1.273 (0.558–2.905)	*p* = 0.567	/	/
Malignancies	1.141 (0.299–4.364)	*p* = 0.847	/	/
DM	1.984 (1.066–3.690)	*p* = 0.031 *	1.638 (0.798–3.361)	*p* = 0.178
Neurology disease	1.059 (0.953–1.178)	*p* = 0.287	/	/
Kidney disease	0.599 (0.245–1.462)	*p* = 0.260	/	/
Leukopenia	2.279 (1.116–4.653)	*p* = 0.024 *	2.659 (1.222–5.787)	*p* = 0.014 *
Lymphocytes	1.805 (0.797–1.478)	*p* = 0.604	/	/
Platelets	0.999 (0.997–1.002)	*p* = 0.625	/	/
Hemoglobin	1.000 (0.991–1.010)	*p* = 0.934	/	/
Ferritin	0.539 (0.312–0.929)	*p* = 0.026 *	0.686 (0.377–1.249)	*p* = 0.211

* Statistically significant.

**Table 6 microorganisms-12-00027-t006:** Predictors of elevated liver biochemistry in patients with COVID-19.

Covariate	Univariate Regression Analyses		Multivariate Regression	
	Exp (B) 95%CI	Significance	Exp (B) 95%CI	Exp (B) 95%CI
Age	0.991 (0.965–1.018)	*p* = 0.521	/	/
Sex	0.462 (0.216–0.987)	*p* = 0.046 *	0.559 (0.240–1.304)	*p* = 0.179
Vaccination	0.646 (0.305–1.370)	*p* = 0.255	/	/
GI symptoms	0.166 (0.061–0.448)	*p* = 0.000 *	0.089 (0.014–0.571)	*p* = 0.011 *
Loss of appetite	0.414 (0.192–0.894)	*p* = 0.025 *	0.187 (0.066–0.525)	*p* = 0.001 *
Nausea	0.565 (0.175–1.823)	*p* = 0.339		
Vomiting	1.210 (0.263–5.558)	*p* = 0.807		
Diarrhea	0.432 (0.187–0.996)	*p* = 0.049*	0.186 (0.062–0.557)	*p* = 0.003 *
SpO_2_	0.953 (0.849–1.070)	*p* = 0.412	/	/
O_2_/L	1.018 (0.956–1.085)	*p* = 0.570	/	/
Comorbidity	0.689 (0.267–1.778)	*p* = 0.441	/	/
CVD	0.813 (0.359–1.842)	*p* = 0.621	/	/
Respiratory disease	0.337 (0.127–0.893)	*p* = 0.029 *	0.334 (0.114–0.980)	*p* = 0.049 *
Malignancy	0.500 (0.099–2.533)	*p* = 0.402	/	/
DM	0.875 (0.364–2.101)	*p* = 0.765	/	/
Neurologic disease	0.515 (0.176–1.507)	*p* = 0.226	/	/
Kidney disease	0.632 (0.198–2.019)	*p* = 0.439	/	/
Leukocytes	0.994 (0.907–1.089)	*p* = 0.895	/	/
Lymphocytes	0.932 (0.631–1.377)	*p* = 0.722	/	/
Platelets	1.000 (0.997–1.004)	*p* = 0.805	/	/
Hemoglobin	1.010 (0.995–1.025)	*p* = 0.183	/	/
Ferritin	1.000 (0.999–1.001)	*p* = 0.260	/	/

* Statistically significant.

## Data Availability

Data are contained within the article.

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
