# Peer review of "Gastrointestinal and Hepatological Manifestations in Severe Acute Respiratory Syndrome Coronavirus 2 Infection: Results from the Major COVID Hospital in Serbia"

_microorganisms, 2023, doi:10.3390/microorganisms12010027_

Round 1

Reviewer 1 Report

Comments and Suggestions for Authors

Comments on the Quality of English Language

See comments to authors

Author Response

Compliments to the authors for their work on this important subject. The manuscript is overall clear and well written. I would like to make some observations that I think require clarification and revision. 

  1. Firstly, I would suggest the inclusion of further literature on GI involvement in COVID-19 patients of European origin, as there might be important population differences in clinical presentation. Particularly, Milano et al. (https://doi.org/10.1177/17562848221104610) reported somewhat similar results recently, which should be discussed in the paper.

Answer: 

  1. Thank you very much for your kind words and suggestions.

We included additional literature regarding GI involvement in COVID-19 patients of European origin, particularly from the authors Milano et al.  

  1. Table 10 (line 198) may not be particularly helpful as reported:

- is the "univariant"(=univariate?) column a list of logistic regressions for dependant=GI manifestations with a single covariate (e.g. sex)? If so, they are significant predictors, but not independent, as they would not be weighted. Presumably, independent predictors (lines 200, 255) would be those in the column "multivariant"(=multivariate?). 

Answer: 

We corrected Table 10. We accept your suggestion: univariant change with univariate, multivariant with multivariate. We assigned columns with covariate.  

- was the multivariate model constructed from the significant predictors of the univariate analyses? 

Answer: 

Yes. We explained multivariate model in section Statistical analysis.  

- the interpretation of some variables depends on how they were coded by the authors for analysis in SPSS but are not clear to the reader. For instance "sex" (ExpB 1.725 for males vs females or vice-versa?) or "leukocytes" (absolute count or high/normal/low?). Although this is clearer in the discussion section, it should be clarified in the table also. 

Answer: 

We accept your suggestion and make changes to the table. 

- the predictor “comorbidity” probably is structurally collinear with both “CVD” and “DM”, as it likely contains both of the latter parameters in its calculation. It may be useful to include “CVD” and “DM” by themselves in the multivariate model. 

Answer: 

We accept your suggestion and make changes to the table. In multivariate models we include only CVD and DM 

Similar considerations for table 11 (line 205). Also the predictor “GI symptoms” probably is structurally collinear with both “loss of appetite” and “diarrhea”, see previous argument. 

Answer: 

We accept your suggestion and make changes to the table. In multivariate models we include only loss of appetite and diarrhea. 

Line 201: among significant predictors, CVD should be included based on reported results (table 10). 

Lines 270-275: ALT/AST serum increases are discussed in the context of conditions related to older age, such as polypharmacy or metabolic disease; however, in this study age was not an overall predictor of liver biochemistry impairment and table 5 suggests a more complex pattern, where ages 45-65 were significantly correlated to increased ALT titers (maybe a nonlinear association?). Some of these observations should be included in the discussion. 

  1. Minor language revisions needed, eg. nephrology disease=kidney disease, platets=platelets (table 10).

Answer: 

  1. We changed “nephrology disease” in to the kidney disease and “platets” in to platelets and we further did spell check for the whole manuscript.

Reviewer 2 Report

Comments and Suggestions for Authors

The manuscript titled "Gastrointestinal and hematological manifestations in Severe Acute Respiratory Syndrome CoV 2 Infection:...." examined data from patients in Serbia that had COVID to better understand the prevalence of GI symptoms in conjunction with the COVID related respiratory symptoms. Overall, the manuscript provides interesting statistics. The figures/tables are clear and organized. And the text is easy to read and accessible to the general audience. I have only minor comments to share:

Font sizes vary throughout the manuscript. Please correct.

In Methods and Materials, how did the authors select the excluding criteria? Please provide information on why the listed excluding criteria were selected over other GI-related diseases.

In the discussion, the authors state that severity of COVID was reduced in patients with GI manifestations when compared to those without GI manifestations. This is an interesting finding. Do the authors have a hypothesis on why that is? It would benefit the manuscript for the authors to add their perspective to that finding. 

Author Response

  1. Font sizes has been corrected according to the Microorganisms MDPI submission guidelines and style templates and we believe that the present manuscript meets MDPI’s style requirements.
  2. In Methods and Materials the authors selected patients with previous known GI disorders (like inflammatory bowel disease etc.), and previous known chronic liver disease (like chronic hepatitis or liver cirrhosis of any etiology), in order to exclude the risks that GI symptoms or abnormal liver biochemistry in COVID19 patients is due to underline chronic disease. The explanation is given in Mtehods and Materials. 
  3. In the discussion, we gave a potential explanation regarding GI manifestations and severity of COVID-19.

Reviewer 3 Report

Comments and Suggestions for Authors

Please, add the id number for the ethical approval.

Please, provide a reason for why the specific time frame from December 1 to December 15 was chosen.

no information regarding sample size calculation is reported. please add

in the statistical analysis section, no information regarding univariate and multivariate regression models are provided. please add.

please, add information regarding adjustments if any. otherwise, add reasons for not considering adjustment.

no information regarding the setting is provided. please add

inclusion/exclusion criteria are not well described. please, be more specific and precise.

there are too many tables. please, consider incorporating table 4 in Table 1

tables 7-8-9 can be moved in supplementary materials

Can you motivate lines 312-314? Could you add adjustment in order to address this limitation?

Please, there are many other limitations including study design. please add.

In the discussion, please add implications of your results in terms of policies and clinical practices.

Author Response

Answers:  

1.The number of Ethical approval is added in main text No 808/7.  

  1. In Methods and Materials the authors explained the study design in more details. 

Following the Reviewer’s and Editor’s suggestions, the statistics used in the manuscript has been checked by the qualified statistician (dr Biljana Milicic, Professor of Medical Statistics and Informatics at the Faculty of Dental Medicine, University of Belgrade). 

We thank the reviewer for their insightful comment. 

In the present study, we found a Gastrointestinal symptoms (at least one) rate of 25.7%. Given a binary outcome occurring in 25,7% of patients, at least 380 participants are necessary to achieve a confidence interval of 0.2 to 0.5 for the overall outcome proportion, at a two-tailed significance level of 0.05 and 0.8 power of study. Additionally, we enrolled 15% more patients to mitigate potential reductions in sample size or protocol deviations. This precaution ensured that our study remained robust and the findings were credible, but unfortunately, only for overall analysis. 

Descriptive statistics were calculated for demographic characteristics and other followed parameters and were presented as frequencies and proportions for discrete measures. All our data were categorical and analysed using Pearson chi-square test. For statistical analysis differences between groups with and without gastrointestinal symptoms, or with and without elevated liver chemistry,  univariable and multivariable logistic regression methods were used. All test variables with a statistical significance of p<0.05 in the univariable model were included in the multivariable model. Signifcant parameters in the multivariate model were considered predictors of diference in primary outcome variables: gastrointestinal symptoms, and elevated liver chemistry. The established level of signifcance was 0.05. Statistical analysis was performed using the IBP SPSS Statistics v28 (Statistical Package for Social Sciences, SPSS Inc, Chicago, Illinois). 

  1. Table 1 and Table 2 are merged in Table 1.

Table 3 and Table 4 are merged in Table 2 

  1. Tables 7, 8 and 9 are moved to supplementary materials as supplementary Tables 1, 2 and 3.

  1. In the part Disscusion we gave the limitations regarding study design and the potential explanation regarding GI manifestations and reduced severety of COVID19 including more details regarding lines 312-314.

  1. We added potential implications of our results in terms of polices,  future research regarding COVID-19 and clinical practice. Consequently, two new references were added to the revised manuscript.

Round 2

Reviewer 3 Report

Comments and Suggestions for Authors

Some of my previous comments have not been taken into account.

Please, revise accordingly: 

Please, provide a reason for why the specific time frame from December 1 to December 15 was chosen.

please, add information regarding adjustments if any. otherwise, add reasons for not considering adjustment.

no information regarding the setting is provided. please add

inclusion/exclusion criteria are not well described. please, be more specific and precise.

Can you motivate lines 312-314? Could you add adjustment in order to address this limitation?

Author Response

Thank you for your comments and suggestions. 

Regarding the time frame, we conducted the study from November 1 to December 15, it was the period when some of the authors were in charge of Covid Hospital Batajnica.  It was easier that way to get all the relevant information regarding the patients included in the study. This is a cross sectional study and because of that there were not any specific reason regarding the period that was taken. However, it was a peak of  COVID19 pandemic in Serbia with the most severe patients coming to a largest COVID Hospital in Serbia at that time. 

Regarding the study design and setting, we changed the whole part of Material and Methods with subtitle “Study design and data source” being more specific and precise. Also we underlined some limitations of our study regarding the study design in the part of discussion (lines 409-414).  

We corrected statistical analyses and showed it in Results, and we added explanation regarding adjustments in the part of discussion, and also in the part regarding limitations of the study (lines 349- 368).